# Thyroid Nodule Characterization: How to Assess the Malignancy Risk. Update of the Literature

**DOI:** 10.3390/diagnostics11081374

**Published:** 2021-07-30

**Authors:** Daniele Fresilli, Emanuele David, Patrizia Pacini, Giovanni Del Gaudio, Vincenzo Dolcetti, Giuseppe Tiziano Lucarelli, Nicola Di Leo, Maria Irene Bellini, Vito D’Andrea, Salvatore Sorrenti, Domenico Mascagni, Marco Biffoni, Cosimo Durante, Giorgio Grani, Giuseppe De Vincentis, Vito Cantisani

**Affiliations:** 1Department of Radiological Sciences, Oncology and Pathology, Policlinico Umberto I, Sapienza University of Rome, 00161 Rome, Italy; daniele.fresilli@hotmail.it (D.F.); patry.shepsut91@gmail.com (P.P.); g.d.gaudio@gmail.com (G.D.G.); vincenzodolcetti@gmail.com (V.D.); gtlucarelli@gmail.com (G.T.L.); nicola.dileo130287@gmail.com (N.D.L.); giuseppe.devincentis@uniroma1.it (G.D.V.); 2Radiological Sciences, Radiology Unit, Papardo-Hospital, 98158 Messina, Italy; david.emanuele@yahoo.it; 3Department of Surgical Sciences, Sapienza University, 00161 Rome, Italy; m.irene.bellini@gmail.com (M.I.B.); vito.dandrea@uniroma1.it (V.D.); salvatore.sorrenti@uniroma1.it (S.S.); domenico.mascagni@uniroma1.it (D.M.); marco.biffoni@uniroma1.it (M.B.); 4Department of Translational and Precision Medicine, Sapienza University of Rome, 00185 Rome, Italy; cosimo.durante@uniroma1.it (C.D.); giorgio.grani@uniroma1.it (G.G.)

**Keywords:** MPUS, thyroid nodule, ultrasound, US-elastography, CEUS

## Abstract

Ultrasound (US) is the first imaging modality for thyroid parenchyma evaluation. In the last decades, the role of ultrasound has been improved with the introduction of new US software, such as contrast-enhanced ultrasound (CEUS) and US-elastography (USE). USE is nowadays recognized as an essential part of the multiparametric ultrasound (MPUS) examination, in particular for the indeterminate thyroid nodule with possible fine-needle aspiration cytology (FNAC) number reduction; even if further and larger studies are needed to validate it. More controversial is the role of CEUS in thyroid evaluation, due to its high variability in sensitivity and specificity. Semi-automatic US systems based on the computer-aided diagnosis (CAD) system are producing interesting results, especially as an aid to less experienced operators. New knowledge on the molecular mechanisms involved in thyroid cancer is allowing practitioners to identify new genomic thyroid markers that could reduce the number of “diagnostic” thyroidectomies. We have therefore drawn up an updated representation of the current evidence in the literature for thyroid nodule multiparametric ultrasound (MPUS) evaluation with particular regard to USE, the US CAD system and CEUS.

## 1. Introduction

The incidence of thyroid cancer is increasing rapidly worldwide, although its mortality rate remains flat [1,2]. Thyroid nodules occur commonly in the general population, mostly as incidental findings, with a prevalence of 19–68% at ultrasound (US) evaluation [3,4], which is the first examination method used for neck imaging.

However, although US is an established and very sensitive method for detecting thyroid nodules, and superior to clinical physical palpation [5,6], it has a relatively low diagnostic performance when it comes to differentiating between benign and malignant nodules.

In fact, when using US, suspicious features such as microcalcifications, marked hypoechogenicity, taller-than-wide shape, and thick irregular or lobulated margins are correlated with malignancy but without high predictive value [7,8,9].

Therefore, when a patient presents with normal thyroid-stimulating hormone levels but US shows suspicious signs [7], a fine-needle aspiration biopsy (FNAB) is advised, which increases the number of cytological examinations that must be performed and the associated health system costs along with it. Furthermore, fine-needle aspiration (FNA) is not a conclusive diagnostic method in all cases, considering its specificity, which ranges from 60 to 98%, and its variable sensitivity, which ranges from 54 to 90%, as well as the frequent eventual clinical need to repeat the test [10,11,12,13,14,15,16].

However, in consideration of the financial burden on health services, and to avoid unnecessary anxiety for patients, it is unrealistic to biopsy every thyroid nodule to obtain histological diagnosis [17]. Thus, an accurate and precise imaging, which nowadays relies mainly on instrumental (US) and biological (FNA) analysis, is warranted.

In the last ten years, there have been a variety of efforts to mitigate the role of overdiagnosis in the increasing incidence of thyroid cancer, ranging from public education programs to guideline and diagnostic changes.

Furthermore, the American Thyroid Association (ATA) guidelines released in 2015 recommended a risk-stratified approach to the utilization of fine-needle aspiration biopsy (FNAB) on nodules, and recommended that those of <1 cm in size should generally not undergo biopsy [7].

In recent years, several studies have proposed the potential use of new ultrasound techniques such as CEUS (contrast-enhanced ultrasound) and, especially, USE (US-elastography) to increase the accuracy of baseline US [18,19] (Figure 1a–d and Figure 2a–e) (Table 1). CEUS was postulated as a means to accurately evaluate the dynamic qualitative and quantitative intensity of tumor perfusion and vascularity, which could be useful for identifying malignant tumors [18].

Moreover, ultrasound elastography (USE) is emerging as a promising additional tool to discern malignant thyroid nodules, allowing increased diagnostic accuracy, especially in comparison with TIRADS (Thyroid Imaging Reporting and Data Systems) [8,17,20]. A firm or hard nodule consistency at palpation is associated with a high risk of malignancy [21]. Due to its ability to assess stiffness as an indicator of malignancy, USE has recently become an additional tool for thyroid nodule differentiation, in combination with conventional US and FNA [21]. In particular, strain ratio elastography (SRE) has shown high sensitivity and specificity, leading the EFSUMB (European Federation of Societies for Ultrasound in Medicine and Biology) to recommend that it should be part of the work-up of thyroid nodule characterization [22].

The role of ultrasound in the evaluation of thyroid nodules is not limited to diagnosis, but also encompasses the evaluation of pre-operative prognostic features, such as extrathyroidal extension [23] and the detection of cervical adenopathies. It is also used to guide interventional procedures [24,25].

Therefore, based on evidence in the literature, we describe the current role of US, TIRADS, CEUS, USE, Artificial Intelligence (AI) and molecular tests for the characterization of thyroid nodules.

## 2. Material and Methods

Four operators (one radiologist and three radiology residents of our institution) carried out research on the literature from the PubMed, Cochrane, EMBASE, Web of Science and Scopus databases from the earliest available article (1998) up to June 2021.

The search terms were sought within the titles and abstract, and were as follows: “thyroid”, followed by each technique included in MPUS evaluation, both written out in full and as abbreviations (ultrasound, US, B-mode, grayscale, color–Doppler, CD, CDUS, contrast-enhanced ultrasound, CEUS, elastography, elastosonography, USE, shear wave, SWE, strain elastography, SE, SRE, CAD-system, Artificial Intelligence, AI).

From the list of all acquired papers, a group of three operators (one radiologist and two radiology residents in our institution) selected the eligible papers, by eliminating duplicates and those which were not pertinent. Those selected for exclusion were commentaries, oral presentations, posters, letters to the editor and any papers not written in English. A similar approach was followed for molecular testing (Pubmed query: “thyroid nodule”[tiab] AND “molecular testing”[tiab]) AND ((“2017/01/01”[PDAT]: “2021/12/31”[PDAT]) AND English[lang]).

## 3. Tirads Lexicon

Ultrasound examination is the preferred first-line imaging modality for evaluating thyroid nodules, due to its non-invasiveness, low cost, wide availability and the abundant information it provides on the characteristics associated with malignancy, included in the Thyroid Imaging Reporting and Data Systems (TIRADS) lexicon: echogenicity, margins and the presence of microcalcifications [26].

Several guidelines indicate how to stratify the ‘risk-of-malignancy’ with US and to follow appropriate diagnostic algorithms. Several studies have recently proposed the TIRADS lexicon as a tool for uniform reporting and consistent evaluation [8,17,20,27].

The TIRADS lexicon is based on internal components (solid, mixed or cystic), echogenicity (hyper-, iso-, hypoechoic or markedly hypoechoic), margins (regular, microlobulated; irregular/spiculated), the presence of calcifications (micro or macro), and shape (taller-than-wide shape is recorded when the anteroposterior diameter is greater than the transverse one [28]; wider-than-tall shape in cases where the opposite is true) at ultrasound evaluation.

The main advantages deriving from the daily use of TIRADS include greater accuracy in the identification of suspicious thyroid nodules; the ability to identify benign nodules that do not require needle aspiration; and the creation of a standardized and reproducible report with a reduced need for FNA [29]. In particular, ACR TIRADS has been found to reduce biopsy rates more than all other TIRADS considered [29,30].

Unfortunately, as shown by an Italian survey, TIRADS are not commonly used and are reported in 27.2% Italian publications [31], primarily by non-radiologist physicians [32], perhaps due to the risk of incorrect classification, laziness in reporting, the many different TIRADS there are to choose from, or poor knowledge of TIRADS.

## 4. Color-Doppler

Many studies have clarified the limited role of CDUS (color–Doppler US) for thyroid cancer diagnosis [33,34,35]. In fact, in 2010, Cantisani et al. retrospectively demonstrated the high frequency of benignity (about 56.7%) in a series of 1090 thyroid nodules with peri- and intra-nodular vascularization at CDUS (pattern III) [34]. Thus, it was not possible to identify a statistically valid correlation between malignancy and pattern III. As a consequence of the above-mentioned limitations, color–Doppler features were not included as a valid sign in the various main TIRADS scores [8,17,20].

## 5. Fine-Needle Aspiration (FNA)

Since US does not yet provide enough accuracy for predicting thyroid malignancy, FNA remains a final minimally invasive diagnostic method to stratify the risk of thyroid malignancy with good accuracy [36].

Fine-needle aspiration cytology (FNAC) still represents the gold-standard technique for thyroid nodule classification, due to its high specificity (60–98%); however, the variable reported sensitivity (ranging between 54 and 90%) should be taken in consideration [10,11,12,13,14].

In the current literature, the main indications for thyroid nodule FNA are:nodule size >1 cm with ≥2 suspicious US malignant featuresall nodules, independent of size, with extracapsular extension or indeterminate/suspicious cervical lymph nodesall nodules, independent of size, in patients who have previously undergone neck radiationa family history of well-differentiated thyroid carcinoma (>2 first-degree relatives)suspicious medullary thyroid carcinoma or a patient affected by multiple endocrine neoplasia (MEN) type 2elevated blood calcitonin levels

The best way to perform a thyroid nodule FNA is under US guidance, documenting the correct needle position inside the nodule and then in aspiration.

Generally, the needle gauges used are approximately 22 to 23, since some papers showed no significant results when using larger-gauge needles, which, conversely, slightly increased the risk of bleeding and the rate of hemorrhagic smears [37].

However, some complications may occur, such as pain and bleeding. The main FNA contraindications are bleeding diathesis, a lack of cooperation by the patient, and in cases where the cytological result will have no therapeutic consequences [38,39,40].

## 6. Multiparametric Ultrasound

Although TIRADS, based mainly on the gray-scale, can be a valid help, due to the fact that there is no single suspicious gray-scale US feature that is sufficiently sensitive or specific for diagnosing malignancy, a considerable overlap and therefore variable accuracies have been reported in literature [41,42,43,44].

Therefore, FNA remains a final, minimally invasive diagnostic method. However, FNA also suffers from limitations related to size and cell sampling, with possible non-diagnostic results (TIR 1), or related to the difficulty in distinguishing between follicular adenoma vs. carcinoma, as in the cytologically indeterminate nodule (TIR 3) in which “microfollicular architectural patterns” may be found, not only in non-neoplastic lesions such as adenomatous hyperplasia, but also in benign lesions such as follicular adenoma, or in malignant ones such as follicular carcinoma, suggesting a need for other less invasive methods [7,36].

There has been much interest, to date, in finding appropriate diagnostic algorithms to stratify the malignancy risk.

To date, fine-needle aspiration biopsy (FNAB) is generally required for nodules >10 mm with suspicious US signs or those with indeterminate signs that are >15 mm [22].

However, it should be taken in account that cytology itself has a reported specificity of 60–98% and a highly variable sensitivity (54–90%), but with considerable non-diagnostic reports [10,11,12,13,14].

To overcome the above-mentioned limitations, multiparametric ultrasound based mainly on USE has recently been proposed as a promising additional tool, due to its ability to evaluate the increased stiffness in a thyroid nodule as a sign of malignancy [45,46].

In fact, in recent years, WFUMB (World Federation for Ultrasound in Medicine and Biology) [47] and EFSUMB (European Federation of Societies for Ultrasound in Medicine and Biology) [48] guidelines reported on the technical details, advantages, limitations and recommendations for SRE and quantitative 2D shear-wave elastography (SWE) [22].

USE is an additional method of ultrasound evaluation that can be performed with qualitative and semiquantitative SRE and quantitative SWE software (in kPa or m/s).

This technique may increase the sensitivity of conventional US alone, and also helps avoid unnecessary invasive examinations, complementing FNAC especially when cytology is non-diagnostic (TIR 1) or indeterminate (TIR 3) [49]. In such cases, SRE seems to be more accurate than SWE [22].

For SRE USE assessment, the operator superimposes the transducer perpendicularly in transverse and longitudinal scans over the neck and then performs periodic free-hand compression cycles, assisted by real-time qualitative indicators that are present in modern equipment. As a consequence, the nodule and the thyroid parenchyma are colored according to their stiffness. In these polychromatic maps, or elastograms, the suspicious hard nodules (those with the lowest or no elastic strain) are displayed in either red or blue, while the opposite color is used to indicate soft nodules (those with higher elastic strain), with intermediate values of stiffness represented via hues of green.

A semiquantitative SRE evaluation may be obtained in several ways. The most widely used method is to superimpose two regions of interest (ROI): the first ROI within the nodule; the second one placed in the adjacent healthy parenchyma. Experts suggest selecting ROIs of similar size and at similar depths. Strain ratio values are automatically calculated by the equipment, dividing the strain of the normal thyroid parenchyma by that of the nodule [22].

For SWE, the transducer is placed perpendicular to the lesion, using mild and controlled pressure to minimize vertical compression artifacts; then an ROI is positioned within the lesion. The equipment, depending on the specific technology used, provides a quantitative evaluation calculated in kPa or m/s [22].

Adequate training, suitable SRE and SWE cut-off values, adequate equipment and clear clinical indications are all necessary to achieve a reliable USE. Experts suggest minimizing pre-compression and carotid pulsation artifacts; avoiding areas with artifacts, gross calcifications, or cystic areas; checking the ROI size and position; and instructing patients on how to cooperate properly [21,46,50]. Future research efforts should focus on reducing inter-observer and intra-observer variability with the use of new software or probes such as quality-indicator tools or 3D US [51,52].

In summary, several papers and meta-analyses have reported that US-elastography is superior to conventional ultrasound.

First, the role of qualitative or semiquantitative USE was addressed in the following studies:−In 2015, Nell et al. published a meta-analysis which included 20 papers with 3908 thyroid nodules assessed by means of qualitative elastosonography using Asteria elastography (ES) classification. The results showed a sensitivity of 85% and specificity of 80% when the threshold elasticity score was between 2 and 3, and a sensitivity of 99% and specificity of 14% when the threshold elasticity score was between 1 and 2. The authors concluded that qualitative elastography has a high ability to detect benign nodules [53].−In 2014, Ghajarzadeh et al. published a metanalysis which included 12 papers with 1180 thyroid nodules (817 benign and 363 malignant) using qualitative US-elastography to differentiate benign and malignant nodules. They achieved a sensitivity of 86% and specificity of 66.7% using a threshold elasticity score between 2 and 3, and a sensitivity of 98.3% and specificity of 19.6% using a threshold elasticity score between 1 and 2. The authors concluded that USE could be a reliable thyroid nodule screening tool [54].

In almost the same years, the better diagnostic abilities of semi-quantitative USE evaluation compared to qualitative USE began to emerge. In particular:−In 2015, in a subgroup analysis, Tian’s metanalysis affirmed the superiority of SR assessment over qualitative USE, with a sensitivity of 86.2% vs. 82.2%, a specificity of 84.8% vs. 80.2% and an area under the curve (AUC) of 0.881 vs. 0.878, respectively [55].−In 2014, Sun et al. published a metanalysis which included 31 papers with 6544 thyroid nodules assessed using real-time ultrasound elastography. They showed that SRE achieved a sensitivity of 85% and a specificity of 80%, whereas elasticity score showed a sensitivity of 79% and a specificity of 77%. The authors concluded that although the diagnostic value of SRE was slightly higher than that of the elasticity score, no significant advantage was demonstrated [56].−In 2013, Razavi et al. published a metanalysis which included 24 papers with 3531 thyroid nodules (2604 benign and 927 malignant) characterized by means of thyroid elastography and B-mode ultrasound. They found higher sensitivity for SRE than for the elasticity score (89% vs. 82%, respectively) but the same specificity of 82% [57].

Subsequently, various meta-analyses have evaluated the diagnostic performance of SWE compared to various gold-standards:−In 2015, Zhan et al. published a meta-analysis which included 16 papers with 2436 thyroid nodules (1691 benign and 745 malignant) examined by means of ARFI (acoustic radiation force impulse ) imaging, in order to evaluate its performance in differentiating between benign and malignant thyroid nodules. They found that ARFI demonstrated a sensitivity of 80% and a specificity of 85%, concluding that SWE could help practitioners identify which patients should undergo surgical treatment [58].−In 2018, Chang et al. published a meta-analysis which included 20 papers with 3397 thyroid nodules, showing a pooled sensitivity of 68% and specificity of 85% for SWE. The authors concluded that SWE is very accurate in distinguishing malignant and benign nodules (AUC 0.9041) [59].−In 2020, Filho et al. published a meta-analysis which included 17 papers with 3806 thyroid nodules (2428 benign and 1378 malignant) evaluated by means of elastosonography, using 2D–SWE equipment produced by various manufacturers. They reported a sensitivity of 77% and a specificity of 76% for T–SWE (Toshiba shear-wave elastography); a sensitivity of 72% and a specificity of 81% for VTIQ (Virtual Touch tissue imaging and Quantification); and a sensitivity of 63% and a specificity of 81% for S-SWE (SuperSonic shear-wave elastography). The authors concluded that 2D–SWE could rule out the malignant nature of thyroid nodules [60].

There are also meta-analyses which compare the two elastosonographic techniques:−In 2017, Hu et al. published a meta-analysis which included 22 randomized controlled trials with 2661 thyroid nodules (2063 benign and 598 malignant) comparing the diagnostic performance of SRE and SWE in characterizing the nature of thyroid nodules. They showed that SRE achieved a sensitivity of 84% and a specificity of 90%, whereas SWE obtained a sensitivity of 79% and a specificity of 87%. The authors concluded that the sensitivity of SRE was superior to that of SWE (0.84 vs. 0.79, respectively) but with comparable values (*p* > 0.05), and that the specificity of SRE was significantly better, statistically speaking, than that of SWE (0.90 vs. 0.87 with *p* < 0.05) [61].−In 2015, Tian et al. published a meta-analysis which included 54 papers with 10,001 thyroid nodules (7380 benign and 2621 malignant) evaluated using the SRE and SWE elastosonography techniques. They reported a sensitivity of 83% and a specificity of 81.2% for SRE, and a sensitivity of 78.7% and a specificity of 80.5% for SWE. The authors concluded that the sensitivity of SRE was better than that of SWE (0.812 vs. 0.787) but they had a comparable specificity [55].

As the aforementioned meta-analyses demonstrate, and the EFSUMB guidelines state, USE techniques are promising diagnostic tools for discriminating thyroid nodules in which the role of the semi-quantitative USE prevails, providing better diagnostic accuracy than the qualitative and quantitative approaches. Unfortunately, SRE suffers from the same limitations as all ultrasound techniques, such as operator-dependence, equipment-dependence and patient-dependence.

Another important issue to be assessed is the potential role of CEUS in thyroid nodule evaluation, since it can provide both a qualitative and quantitative evaluation of the nodule’s contrast enhancement, which could be related to the malignancy risk. To date, there are no unified standards for quantitative or qualitative studies and no single feature of CEUS seems to be sensitive and specific enough for the diagnosis of malignancy [62,63].

In fact, various studies evaluate the diagnostic accuracy of contrast-enhanced ultrasound (CEUS) for distinguishing malignant thyroid nodules from benign ones, but it remains controversial, due to its high variability in sensitivity and specificity [18,62,63].

In 2019, Trimboli published a meta-analysis which included 14 original articles with 1515 thyroid nodules (741 malignant at histology) characterized by means of Sonovue administration. He found that although CEUS achieved good performance in characterizing the thyroid nodule, with a pooled sensitivity of about 85%, a pooled specificity of 82%, a PPV of 83%, and an NPV of 85%, these data were affected by moderate inconsistencies, in particular for specificity (I2 = 71%) and PPV (I2 = 72%), and by publication bias, for sensitivity (*p* = 0.02) and NPV (*p* = 0.01) [62].

In 2018, Liu et al. published a meta-analysis which included 33 papers with 3742 thyroid nodules (1780 malignant nodules) characterized through Sonovue administration. They found a sensitivity of 88% and specificity of 88% in distinguishing the nature of thyroid nodules, concluding that CEUS has good accuracy, but that there was, at the time of publication, no standardization of CEUS procedures, and that no single feature of CEUS was accurate enough to diagnose the nature of thyroid nodules. [63].

To date, there is no consensus regarding the accuracy of CEUS or its ability to improve the diagnostic accuracy of US imaging reporting systems (such as TIRADS) [64].

In fact, the EFSUMB guidelines on non-hepatic CEUS applications state that CEUS cannot be recommended for clinical use in evaluating thyroid nodules, but only for research [18].

In summary, the current literature shows that USE is a promising technique for thyroid nodule characterization, and it seems to be more sensitive than baseline US and CEUS [65]. It is especially recommended as an additional tool alongside conventional US to improve specificity [21], especially with regard to strain technique, and it is also recommended as a tool to monitor lesions previously diagnosed as benign using FNA [18].

### Multiparametric Ultrasound (MPUS) Role for Indeterminate Thyroid Nodule Evaluation

Although FNA is the gold standard for thyroid nodule classification, problems such as inadequate sampling or indeterminate results are frequent. An indeterminate result represents a “gray” diagnostic area, estimated to occur in 5–20% of cytological reports with the presence of cellular atypia of indeterminate significance (TIR3 category) [66]. As a consequence, a significant number of patients receive unnecessary thyroid surgery, more for diagnostic than for therapeutic purposes, leading to increased clinical risks, costs, and a certain level of complications. Therefore, improvement and refinement of noninvasive nodule diagnosis is needed.

TIR3 nodule management is widely debated, given that no more than 30% of nodules will be diagnosed as malignant at histological examination and, in particular, fewer than 10% turn out to be malignant in TIR3A nodules and 15–30% in in the TIR3B nodules [67]. In view of the above, TIR3A nodules are generally candidates for a conservative approach (ultrasound follow-up and FNAC repetition), while the recommended treatment for TIR3B nodules is surgery, with total or partial diagnostic thyroidectomy [15,16]

MPUS provides valid support to define the malignancy risk of indeterminate nodules and reduce the need for surgery for non-malignant diseases. In this regard, our group reported on the possible contribution of SRE in the assessment of thyroid nodules with indeterminate cytology, finding a significant correlation between strain ratio values >2.05 and malignancy risk, with 87.5% sensitivity and 92% specificity [68].

Celletti et al. [49] found a better correlation with the cytological results from strain elastography values (cut-off > 1.96) compared to shear-wave elastography values (*p* < 0.05). Its addition to K-TIRADS, results in increased sensitivity (92.9% vs. 71.4%) and negative predictive value (NPV) (96.3% vs. 87.5%). These observations are mostly confirmed in nodules larger than one centimeter (AUC value: 0.881 vs. 0.780), although the greater NPV value was also found in nodules ≤1 cm (near 100%).

In this regard, in 2019, Qiu et al. published a meta-analysis which included 20 studies with 1734 indeterminate thyroid nodules (25.9% rate of malignancy), evaluating the use of elastography to differentiate between benign and malignant indeterminate thyroid nodules. They showed an overall USE sensitivity of 76.6% and a specificity of 86.7%; an RTE (real-time elastography) sensitivity of 71.5% and a specificity of 85.3%; an SRE sensitivity of 81.3% and a specificity of 89.4%; and an SWE sensitivity of 83.8% and a specificity of 87.2%. They concluded that both strain and shear-wave elastography produced good results and they could be combined with conventional US in the coming years [69].

In addition, Samir et al. [70] concluded that the quantitative approach (SWE) is a valuable tool for assessing pre-operative malignancy risk in thyroid nodules with indeterminate cytology, showing a sensitivity of 82% and a specificity of 88%, with a 22.3 kPa cut-off.

Conversely, Bardet et al. concluded that the SWE cut-off values could not improve the differentiation of benign and malignant tumors among thyroid nodules with indeterminate cytology [71].

## 7. Computer-Aided Diagnosis (CAD) System

To overcome limits of inter-observer variability and limit the time needed for tests, several classification systems that combine various US findings have been developed to estimate the likelihood of malignancy and select nodules for FNA biopsies. The application of these systems, which are endorsed by international scientific societies [20,72,73], has been proven to reduce inter-observer variability, although further improvements are desirable. To this end, using artificial intelligence, CAD may be able to discern malignant nodules from benign ones with a rate of accuracy similar to that of expert radiologists [74,75,76,77,78,79] and may also contribute to reduce intra- and inter-observer variability.

The main advantages deriving from the use of a CAD system use include the quick and easy application of TIRADS systems, a reduction in inter-observational variability, and improvement in diagnostic accuracy, with the CAD system operating as a “second eye” for the localization, identification and evaluation of malignant lesions.

In the literature, some CAD systems have been reported which automatically define the shape, composition, echogenicity and margins of nodules. Other features, such as the presence of calcifications, stiffness according to USE, and color–Doppler vascular pattern, have to be input manually. Fresilli et al. showed that S-Detect™ does not provide any clinical advantage to the expert clinician, but it may be a useful tool for less experienced operators for training purposes, as an aid in recognizing suspicious US features and expediting learning of the TIRADS scoring process and its practical application [79].

However, there are also two meta-analyses which evaluate the performance of computer-aided diagnosis systems in the US evaluation of thyroid nodules:−In 2020, Xu et al. published a meta-analysis which included 19 papers with 4781 thyroid nodules, analyzing the performance of CAD systems in diagnosing malignant thyroid nodules. Classic machine learning showed a sensitivity of 86% and a specificity of 85%, and the deep learning-based CAD system showed a sensitivity of 87% and a specificity of 85%. The authors concluded that CAD systems could help in diagnosing malignant thyroid nodules, but that experienced radiologists may be superior to CAD systems, especially for real-time diagnosis [80].−In 2019, Zhao et al. published a meta-analysis which included 5 papers with 723 thyroid nodules, analyzing the performance of CAD systems in diagnosing malignant thyroid nodules. They found a sensitivity of 87% and a specificity of 79%, concluding that the CAD system had a similar sensitivity to experienced radiologists when it came to the evaluation of thyroid nodules, but with a lower specificity [81].

## 8. Molecular Diagnostics

In the last few decades, the availability of new genomic technologies has improved our knowledge about the molecular mechanisms involved in thyroid cancer [82] and breast cancer [83] (Table 1). Pathogenesis usually involves dysregulation of the mitogen-activated protein kinase (MAPK) and phosphatidylinositol-3 kinase (PI3K)/AKT signaling pathways. The first pathway is crucial for PTC initiation (e.g., mutations of the BRAF and RAS genes or gene fusions of RET/PTC and TRK); the second pathway is thought to be critical in FTC initiation (e.g., mutations in RAS, PIK3CA, and AKT1, inactivation of PTEN). Mutations in RET proto-oncogene account for most medullary thyroid cancer cases (sporadically or as inherited germline events). This knowledge paved the way for the clinical application of the molecular testing of FNAB samples. This is the most recent approach to a better stratification of the malignancy risk of thyroid nodules with indeterminate cytology [84], and mainly aims to reduce the number of “diagnostic” surgeries.

The first attempts included searching for a small number of somatic mutations (mainly of BRAF and RAS genes). Nowadays, commercial tests use panels of multiple molecular alteration, such as somatic mutations, gene expression and microRNA (miRNA). Current ThyroSeq (version 3) includes the targeted next-generation sequencing analysis of 112 genes for gene fusions, mutations, copy-number alterations and abnormal gene expression [85]. The current version of the Afirma Genomic Sequencing Classifier (GSC) test includes 12 classifiers composed of 10,196 genes (RNA sequencing approach) [86]. Overall, both ThyroSeq and Afirma assays currently have NPVs that make them useful in ruling out malignancy. The diagnostic performances of these tests were compared in a recent clinical trial. The authors reported no significant differences in sensitivity (97% vs. 100%) or specificity (85% vs. 80%); up to 49% of patients with indeterminate nodules were able to avoid diagnostic surgery [87].

Other techniques are based on the evaluation of levels of miRNA – small, noncoding molecules able to interact with multiple pathways [88]. The performance of a smaller mutation panel combined with an miRNA risk classifier (ThyGeNEXT + ThyraMIR) was recently validated in a multicenter study [89], which reported good sensitivity (95%) and specificity (90%) for malignancy in indeterminate nodules. Some other, potentially less expensive, molecular tests were recently proposed, but need prospective and multicenter validation; these include a dual-component molecular assay (an extended DNA and RNA mutation panel plus a single miRNA expression level) [90], and the analysis of selected long non-coding RNAs (MALAT1, HOTAIR, PVT1) [91].

While molecular tests have been proven to reduce the number of surgeries in vali-dation studies, the applicability of these approaches needs to be verified in different settings [92]. The real performance in clinical practice varies according to the pretest probability of malignancy (i.e., the disease prevalence in the given population). In fact, while sensitivity and specificity are intrinsic properties of the test, predictive values vary: the PPV increases and the NPV decreases when the malignancy rate increases. Highly suspicious nodules, according to their US features, should be referred for treatment: the pretest risk of malignancy is so high that molecular tests may not have an adequate NPV in this case. Furthermore, there is an association between some US features and mutations: for example, microcalcifications, non-parallel orientation, spiculated/microlobulated margins and hypoechogenicity [93], higher American College of Radiology scores [94], size after CEUS enhancement [95], no CEUS enhancement [96], and longer time to peak enhancement [97] may predict the BRAF mutational status. Also, the rate of nodules with non-parallel orientation, spiculated/microlobulated margins, and hypoechogenicity linearly increases in nodules with a BRAF mutation, and with both TERT and BRAF mutations [98]. According to other reports, US features alone may not adequately predict mutational status [99].

For these reasons, several models have been proposed for the direct integration of molecular and sonographic data [100,101,102]. These models have yet to be validated in multicenter and prospective studies.

## 9. Conclusions

In conclusion, many papers have already demonstrated the high diagnostic performance of USE, and especially of semi-quantitative USE, in the characterization of thyroid nodule, noting several advantages, such as low cost and speed, compared to using the same probe and US machine for conventional US. These results are easily repeatable, yielding the low inter-observation and intra-observational variability, and are safe to use because there is no need for a contrast medium. Use of an US CAD system represents an active research field. It will probably become the main method of the next future; currently, however, the scientific evidence is still weak. The use of CEUS in the evaluation of thyroid nodules has important limitations, such as the intravenous use of a contrast medium and higher costs and examination time; as such, its application in daily clinical practice cannot currently be justified.

Some issues remain open. Which is the most accurate elastographic technique? How much US-elastography is reproducible? What is the potential USE in the evaluation of indeterminate thyroid nodules? To what extent does thyroiditis affect thyroid stiffness values? Can MPUS reduce unnecessary FNAC, treatments, and follow-up exams?

Further large-scale multicenter studies and optimized ultrasound software are advised to improve the accuracy of US in the risk stratification of thyroid nodules.

## Figures and Tables

**Figure 1 diagnostics-11-01374-f001:**
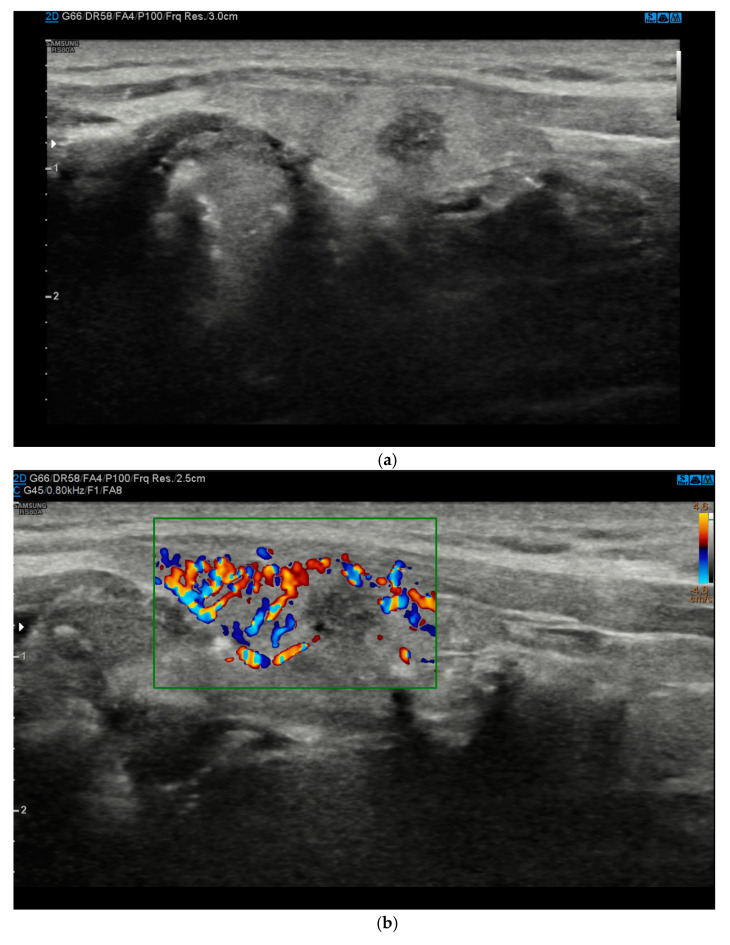
(**a**)**.** At B-mode Ultrasound (US), the lesion appeared round-shaped, hypoechoic with irregular margins (EU-TIRADS 5). (**b**). At color–Doppler US evaluation, the lesion showed no internal or peripheral vascularization (pattern I). (**c**). At US-Elastography (USE) evaluation, the lesion appeared stiff (ECI: 4.03). (**d**). At S-Detect evaluation, the lesion suggested intermediate suspicion of malignancy (TIRADS 4). Finally at histology it was identified as a papillary carcinoma.

**Figure 2 diagnostics-11-01374-f002:**
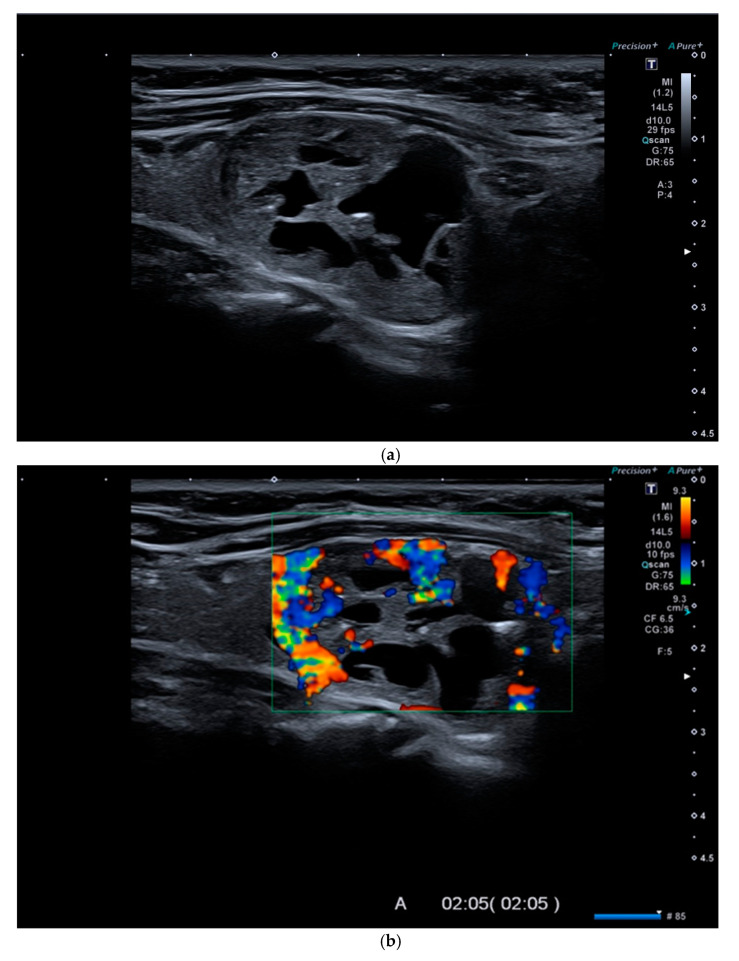
(**a**). At B-mode Ultrasound (US), an oval-shaped nodule with mixed ecostructure, some internal fluid areas and smooth margins, was identified (EU-TIRADS 3). (**b**). At color–Doppler US evaluation, the lesion appeared with internal and peripheral vascularization (pattern III). (**c**). At Strain Ratio Elastography (SRE) evaluation, the lesion appeared soft (SR 1.13). (**d**). At Shear Wave Elastography (SWE) evaluation, the lesion appeared soft (2.24 m/s). (**e**). At CEUS evaluation, the lesion appeared solid and richly vascularized, similar to the surrounding thyroid parenchyma without wash-out. At histology, the lesion was confirmed to be a follicular hyperplasia.

**Table 1 diagnostics-11-01374-t001:** Pros and cons of each MPUS method and molecular test.

	PROS	CONS
**Qualitative USE**	Non-invasive No contrast medium Safe Same probe and US machine of B-mode evaluation Defined in some guidelines Not overly time-consuming Good intra-observed agreement	Subjective evaluation (low inter-observed agreement) Non-univocal classification (4- or 5-point scale scoring systems) Soft carcinoma evaluation (thyroid follicular carcinoma) Undefined thyroiditis influence Calcific nodule may be wrongly assessed Cystic nodule may be wrongly assessed
**Semi-Quantitative USE**	Non-invasive No contrast medium Safe Same probe and US machine of B-mode evaluation Defined in some guidelines Not overly time-consuming Good intra-observed agreement Good inter-observed agreement	Univocal cut-off missing Soft carcinoma evaluation (thyroid follicular carcinoma) Undefined thyroiditis influence Calcific nodule may be wrongly assessed Cystic nodule may be wrongly assessed
**Quantitative USE**	Non-invasive No contrast medium Safe Same probe and US machine of B-mode evaluation Defined in some guidelines Not overly time-consuming Good intra-observed agreement Good inter-observed agreement	Univocal cut-off missing Two different measurement units (m/s and kPa) Soft carcinoma evaluation (thyroid follicular carcinoma) Undefined thyroiditis influence Calcific nodule may be wrongly assessed Cystic nodule may be wrongly assessed
**CEUS qualitative**	Minimally invasive Safe (anaphylactic reactions very rare) Same probe and US machine of B-mode evaluation	Sulfur-based contrast medium use Not approved in clinical practice by the guidelines (only approved for research) Undefined intra-observed agreement Undefined inter-observed agreement More time-consuming than USE
**CEUS quantitative (T/I curve)**	Minimally invasive Safe (anaphylactic reactions very rare) Same probe and US machine of B-mode evaluation	Sulfur-based contrast medium use Not approved in clinical practice by the guidelines (only approved for research) Undefined intra-observed agreement Undefined inter-observed agreement More time-consuming than USE
**US CAD System**	Non-invasive No contrast medium Safe Same probe and US machine of B-mode evaluation Not overly time-consuming Good intraobserved agreement Good interobserved agreement Improves the accuracy of inexperienced or non-specialist radiologists	Not approved in clinical practice by the guidelines (only approved for research) Accuracy not superior to experienced radiologists
**Molecular test**	May be performed on cytological specimens	Cost-effective only in carefully selected nodules Variable performance according to the pre-test probability of malignancy High costs (still unavailable/unaffordable in many countries)

## Data Availability

No new data were generated for this review article.

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
