# Peer review of "Thyroid Nodule Characterization: How to Assess the Malignancy Risk. Update of the Literature"

_diagnostics, 2021, doi:10.3390/diagnostics11081374_

Round 1

Reviewer 1 Report

Congratulations on this well-documented review, having a current topic of interest in medical practice.

I recommend that the abbreviations be explained when they first appear in the text.

  • Abstract - line 17 - if the abbreviation FNAC refers to fine-needle aspiration cytology, it is recommended to render under the form: fine-needle aspiration cytology (FNAC)
  • Same for MPUS, which first appears in row 16
  • TIRADS also appears for the first time in row 6

Although it is a review, I consider it useful to introduce a table to highlight and summarize all the methods described (with the pros and cons of each procedure)

Other interesting new references with reference to cancer that can be introduced if desired:
- DOI: 10.25083/rbl/24.5/813.819
- DOI: 10.22543/7674.71.P3439

Please reformulate the conclusions. These must belong to the authors. There should be no references in the conclusions.

Author Response

Dear Editor thanks a lot for the opportunity to improve our paper entitled "Thyroid Nodule Characterization: How to Assess the Malignancy Risk. Update of the literature" according to the reviewers comments. We tried to do our best to accomplish any requirements. Please find below poit to point answers, we corrected the manuscript following the reviewers comments, requests and queries. Looking forward your positive answer, i send to you my warmest regards. Sincerely. Vito.

Reviewer 1 point 1: "I recommend that the abbreviations be explained when they first appear in the text. Abstract - line 17 - if the abbreviation FNAC refers to fine-needle aspiration cytology, it is recommended to render under the form: fine-needle aspiration cytology (FNAC). Same for MPUS, which first appears in row 16. TIRADS also appears for the first time in row 6"

  • We have explained each abbreviations when they appear in the manuscript as requested. Thank you.

Reviewer 1 point 2: "Although it is a review, I consider it useful to introduce a table to highlight and summarize all the methods described (with the pros and cons of each procedure)

  • Thanks a lot for your suggestion. Indeed we have introduced a table with Pros and Cons of all methods described. 

Reviewer 1 point 3: "Other interesting new references with reference to cancer that can be introduced if desired:
- DOI: 10.25083/rbl/24.5/813.819
- DOI: 10.22543/7674.71.P3439"

  • We have introduced reference with DOI 10.22543/7674.71.P3439. Thank you.

Reviewer 1 point 4: "Please reformulate the conclusions. These must belong to the authors. There should be no references in the conclusions"

  • We reformulated and improved the conclusions. Thank you

Reviewer 2 Report

The authors present a narrative review of the thyroid nodule characterization.

The title 'Multiparametric Ultrasound of Thyroid Nodule Characteriza tion: How to Assess the Malignancy Risk' may be misleading as 'Multiparametric Ultrasound' is a part of the content of the article and is not stated to be a review article.

The objective is not clear. The authors indicate in the introduction 'Therefore, based on the evidence of the Literature, we have provided the current role of US, TIRADS, CEUS, USE, Artificial Intelligence (AI) and molecular test for the thyroid nodule characterization'.  However,  the authors do not indicate the method of bibliographic review followed (keywords, search strategies, article inclusion criteria, exclusion criteria, databases used).

If a systematic review method has not been followed (PRISMA and Cochrane guidelines) it is difficult to establish the criteria of evidence and recommendation to which they refer.

Author Response

Dear Editor thanks a lot for the opportunity to improve our paper entitled "Thyroid Nodule Characterization: How to Assess the Malignancy Risk. Update of the literature" according to the reviewers comments. We tried to do our best to accomplish any requirements. Please find below ponit to point answers. We have corrected the manuscript following the reviewers comments, requests and queries. Looking forward your positive answer, i send to you my warmest regards. Sincerely. Vito.

Reviewer 2): "The title 'Multiparametric Ultrasound of Thyroid Nodule Characterization: How to Assess the Malignancy Risk' may be misleading as 'Multiparametric Ultrasound' is a part of the content of the article and is not stated to be a review article.

  • We have changed title in "Thyroid Nodule Characterization: How to Assess the Malignancy Risk. Update of the literature" in order to introduce the molecular test paragraph without misleadings. Thank you.

Reviewer 2): "the authors do not indicate the method of bibliographic review followed (keywords, search strategies, article inclusion criteria, exclusion criteria, databases used) If a systematic review method has not been followed (PRISMA and Cochrane guidelines) it is difficult to establish the criteria of evidence and recommendation to which they refer."

  • We have added "Material and Methods" paragraph where we have explained our literature research protocol. This paper is a narrative review and not a systematic review. Howevere we tried to make our objective and following explaination clearer.  Thank you.

Round 2

Reviewer 2 Report

Thanks to the authors for considering the comments and making changes to the original manuscript.